# Development and Calibration of a 3D Micromodel for Evaluation of Masonry Infilled RC Frame Structural Vulnerability to Earthquakes

**Filip Anić** [1], **Davorin Penava** [1,*], **Vasilis Sarhosis** [2] **and Lars Abrahamczyk** [3]

[1] Faculty of Civil Engineering and Architecture Osijek, Josip Juraj Strossmayer University of Osijek, Vladimir Prelog 3 Str., HR-31000 Osijek, Croatia; filip.anic@gfos.hr

[2] Faculty of Engineering and Physical Sciences, School of Civil Engineering, University of Leeds, Leeds LS2 9JT, UK; v.sarhosis@leeds.ac.uk

[3] Chair of Advanced Structures, Faculty of Civil Engineering, Bauhaus-Universität Weimar, Marienstr. 7A, 99423 Weimar, Germany; lars.abrahamczyk@uni-weimar.de

\* Correspondence: davorin.penava@gfos.hr

**Abstract:** Within the scope of literature, the influence of openings within the infill walls that are bounded by a reinforced concrete frame and excited by seismic drift forces in both in- and out-of-plane direction is still uncharted. Therefore, a 3D micromodel was developed and calibrated thereafter, to gain more insight in the topic. The micromodels were calibrated against their equivalent physical test specimens of in-plane, out-of-plane drift driven tests on frames with and without infill walls and openings, as well as out-of-plane bend test of masonry walls. Micromodels were rectified based on their behavior and damage states. As a result of the calibration process, it was found that micromodels were sensitive and insensitive to various parameters, regarding the model's behavior and computational stability. It was found that, even within the same material model, some parameters had more effects when attributed to concrete rather than on masonry. Generally, the in-plane behavior of infilled frames was found to be largely governed by the interface material model. The out-of-plane masonry wall simulations were governed by the tensile strength of both the interface and masonry material model. Yet, the out-of-plane drift driven test was governed by the concrete material properties.

**Keywords:** RC frames; unreinforced masonry infill walls; openings; computational micromodel; calibration; sensitivity analysis; in-plane seismic load; out-of-plane seismic load; structural vulnerability

## 1. Introduction

During the seismic action, buildings made of frames with masonry infill walls are excited in both in-plane (IP) and out-of-plane (OoP) directions by inertial and inter-story drift forces. In order to better comprehend the effects of such actions, the field of earthquake engineering studied the effects of those load directions separately and in combination. During the action, the surrounding frames interact with the infill wall causing damage to both [1,2]. The infill wall affects the overall behavior, performance, and failure mechanisms of the system [3]. The interaction is more pronounced in IP than in the OoP field, and this is exaggerated in the case of static and quasi-static methods. IP loading is, by its nature, administered via inter-story drift forces. Contrariwise, most OoP studies were done with inertial, while only three were done with inter-story drift forces [4].

Furthermore, due to the frame–infill wall interaction, it was found that openings had additional effects on the seismic performance of such structures. This is true in the IP studies, where the openings were better and more systematically studied [5–8], unlike the OoP field, where even some contradictions were found [4]. In the IP studies, various opening sizes, positions and detailing were tested. It was found that infill walls even with openings improve stiffness, ductility, energy dissipation, and lateral load resistance when

compared to bare frame specimens; yet, less so than with fully infilled ones. Positions also played a major load, whereas the closer opening to a diagonal strut is a loss of beneficial properties of the infill, this is exaggerated if the opening is located at the loaded corner of the infill. In the case of OoP studies, there were no studies on opening position, area, detailing or tests that include openings with inter-story drift forces. From the literature, it is unclear as to how the openings reduce the OoP load-bearing capacity. However, openings do certainly allow the beneficial, yet limited arching-action development. The limitation is in the reduced ductility, which was consistent with all experiments. Note that there are no studies with combined IP and OoP loads with openings, not with OoP inertial or drift forces.

There are three main computational approaches of modelling the frames with masonry infill walls: macro-; mezzo-; and micro-models. Macromodels are good for modelling whole multi-story buildings and for fast calculations. They are usually made from frame elements, with one or more compression struts that connect frame corners or elements surrounding the corners. Compression struts are used to model IP, while with the addition of point mass, OoP inertial loads can be simulated. Mezzomodels are in between micro- and macro-models. It also provides faster computation; however, its implementation is not as ingrained as the other two. The model usually consists of frame and shell elements. The shell element is used to model the infill wall and the OoP inertial load is applied via area load to the shell. Finally, the micromodel is assembled from multiple macroelements that usually represent each element (e.g., frame, each masonry unit, each rebar, interface elements). The micromodels are usually complex due to possessing numerous parameters and their interaction with other macroelements (e.g., 2D interface or 1D truss element with 3D solid element, etc.). Furthermore, the micromodels are usually time-consuming; hence, they are not favored in common practices. Yet, the micromodels can simulate more effects and provide more insight than the other methods. All three computational models are represented in IP and OoP fields separately. However, in the IP and OoP combination, only the macromodels are predominant, e.g., [9–12]. They have also successfully incorporated the influence of openings in the combined direction by having two-point masses for OoP simulations that are mutually connected by spring element and by struts to the frame [9–11]. It is worth mentioning that there are also novel and innovative approaches to the topic, such as the work of Rossi et al. [13] that combines both the structural response and economic losses.

This paper presents the micromodel development and its calibration. Micromodels represent reinforced concrete (RC) frames with and without infill walls and openings (Figure 1). The models are based on the following experimental series: (1) IP cyclic, quasi-static test series on frames (Figure 2a) with and without infill wall and openings [5]; (2) OoP monotonic bend test series on masonry walls (Figure 2b), with load parallel and perpendicular to bedjoint [14]; and (3) OoP drift driven cyclic, quasi-static test series on frames with and without infill wall and openings (Figure 2c) [15]. All test series were done in the Faculty of Civil Engineering and Architecture Osijek. Additionally, all the test series included the same frames, infill wall units, openings (dimensions and positions), and mortar type as presented in Figure 1 and Table 1. The outcome of the paper is its computational analysis of various parameters from which the governing parameters were found and compared simultaneously in all directions and models. The end goal is to use the calibrated models for combined IP and OoP load direction in order to better understand their interaction.

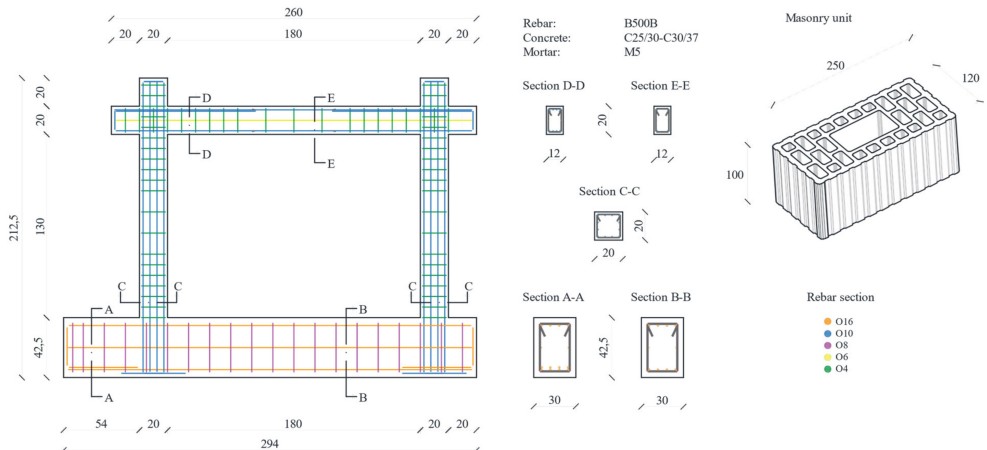

**Figure 1.** Specimen's properties.

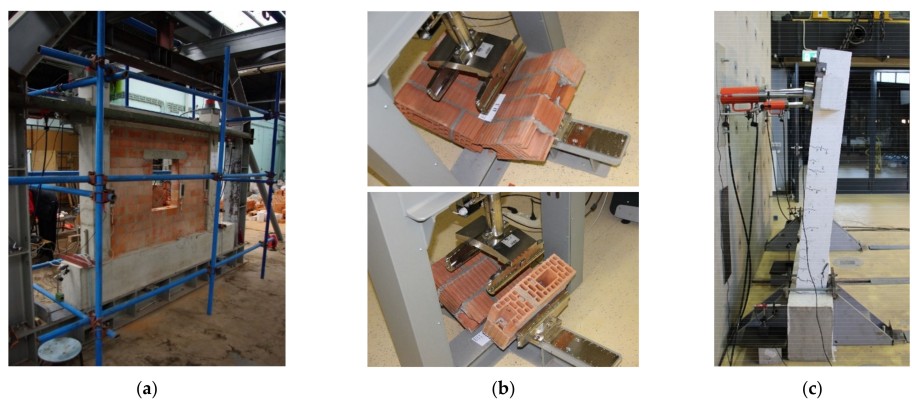

| (a) | (b) | (c) |

**Figure 2.** Photographs of conducted test series. (**a**) IP tests on a frame with infill wall and [5]; (**b**) OoP bend tests on masonry walls [14]; (**c**) OoP tests on frames with infill walls and openings [15].

**Table 1.** Geometrical characteristics of tested specimens.

| Model | | Appearance | $l_o/h_o$ (cm) | $A_o$ (m²) | $A_o/A_i$ (/) | $e_o$ (cm) | P (cm) |
| Name | Mark | | | | | | |
|---|---|---|---|---|---|---|---|
| Bare frame | BF | | / | / | / | / | / |
| Full infill | FI | | / | / | / | / | / |
| Centric door | CD | | 35/90 | 0.32 | 0.13 | $l_i/2 = 90$ | / |
| Centric window | CW | | 50/60 | 0.30 | 0.14 | $l_i/2 = 90$ | 40 |
| Eccentric door | ED | | 35/90 | 0.32 | 0.14 | $h_i/5 + l_o/2 = 44$ | / |
| Eccentric window | EW | | 50/60 | 0.30 | 0.13 | $h_i/5 + l_o/2 = 44$ | 40 |

Where: *l* length, *h* height, *A* area, *e* eccentricity, *P* parapet height, $_i$ infill, and $_o$ opening.

## 2. Micromodel Development

### 2.1. Methods of Computing and Calibrating

To develop and simulate computational micromodels, Atena Engineering 3D software [8] was used. It provides means of nonlinear analysis for concrete and reinforced concrete structures including concrete cracking, crushing, and reinforcement yielding [16].

The software was used to model the FEM structure with its boundary, material, and computational properties (Figure 3). The model was then exported as an *input* file (.inp), with which, the modification was made either by hand or handwritten scripts. Within the input file, more properties could be reached and modified than those available in the standalone program, like the modified Newton–Raphson method.

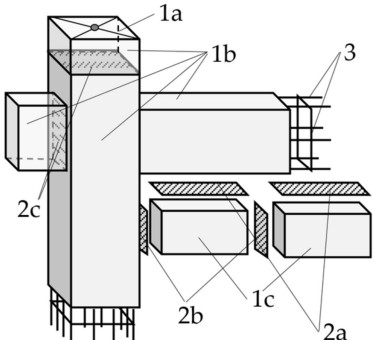

| # | Space | Type | Usage |
|---|---|---|---|
| 1a | 3D | Solid | Plate |
| 1b | 3D | Solid | Concrete |
| 1c | 3D | Solid | Masonry |
| 2a | 2D | Interface | Bedjoint |
| 2b | 2D | Interface | Headjoint |
| 2c | 2D | Interface | Perfect con. |
| 3 | 1D | Truss | Rebar |

**Figure 3.** Parts of the micromodel.

Along with the mentioned, the *modified Newton–Raphson method* was used for stepwise calculations. It was chosen on the simple terms of shortening the time consumption as IP cyclic simulations lasted up to 6 days. The reason behind shortened computing time is that, unlike the full Newton–Raphson method, this one assembles and eliminates stiffness matrix only once. Effectively shortening iteration computing process, while impairing convergence. Note that the computations were initially done with full, and then transferred to the modified method. There were no noticeable differences between the two.

### 2.2. Methodology

The calibration/sensitivity analysis started with IP cyclic, quasi-static tests of frames with masonry infill walls, followed by OoP bending tests of masonry walls, and finally, the OoP cyclic, quasi-static tests on frames with masonry infill walls. The flowchart, i.e., an algorithm of the calibration process is presented in Figure 4. The calibration process started with the BF model, to isolate the behavior of the RC frame. When the BF model was calibrated, it proceeded to the FI model. If there were modifications to the RC frame, the calibrations reverted to the BF point. If there was a change in infill wall properties, it reverted to the FI model. The same pattern translated to models that contained openings. After the IP models were considered calibrated, the calibration moved onto the OoP models. Likewise, if there was a change in the properties of concrete the calibration then fallen back to the IP–BF calibration or IP–FI calibration if there was a change in masonry material.

### 2.3. Micromodel Morphology

In general, the micromodels were built from several macroelements that simulated one or more parts of the specimens. Those parts are presented in Figure 3, where *solid* elements were used to simulate concrete, masonry, and elastic plates. Elastic plates were placed to transmit the load onto the frame while avoiding the numerical singularity. Further on, the *contact*, *interface-gap* elements were used to simulate the connection between solids, while a *perfect connection* was used to bind frame elements and elastic plates into one superstructure. Note that interface-gap elements do not simulate just the mortar, as they also simulate the whole contact that includes the surrounding solids. The 1D truss elements

were used to simulate the rebars. A perfect connection was used between the rebar and the surrounding frame.

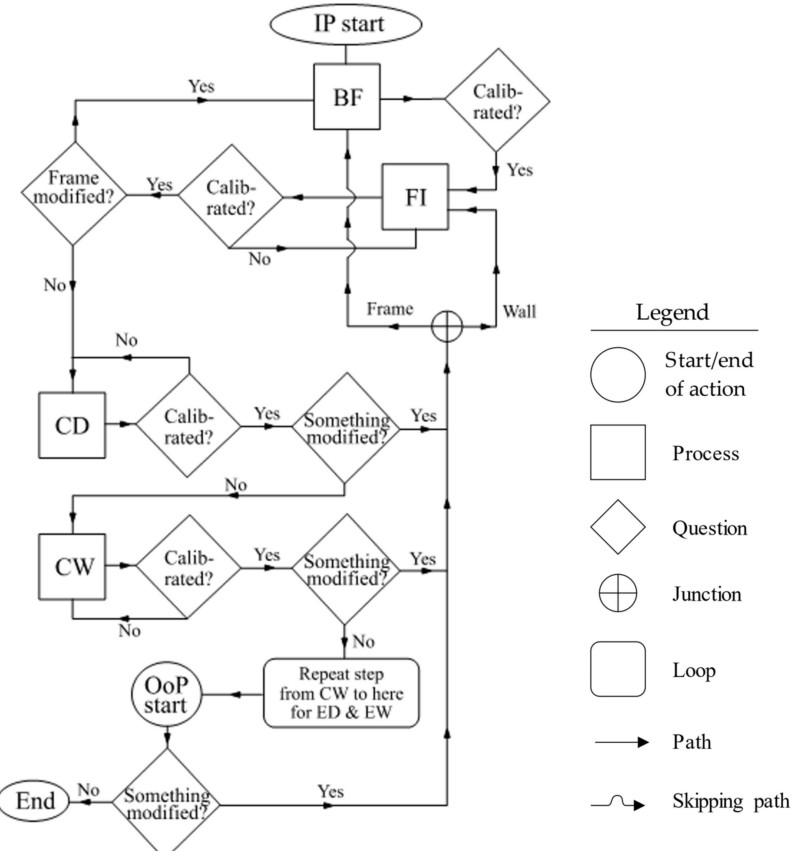

**Figure 4.** Calibration flowchart.

The FEM cube mesh sized of 4 cm was adopted from Penava et al.'s (2016) [17] models. Note that the size was not varied, as smaller sizes would cause prolonged computational times, that extended up to 6 days in case of IP cyclic, quasi-static computations.

In this section, some basic and some more detailed descriptions of material models are presented to clarify their properties and parameter values or ranges that were modified during the calibration.

### 2.4. Material Models

#### 2.4.1. Fracture–Plastic Constitutive Model Material Model

*CC3DNonLinCementitious2* was used to simulate the behavior of both the concrete and masonry. The model combined constitutive models for tensile, i.e., fracturing, and compressive–plastic behavior. The material model is based on orthotropic smeared crack formulation along with the crack band model. Furthermore, it employs the Rankine failure criterion, uses exponential softening with rotated or fixed crack model. The hardening/softening plasticity model is based on the Menétrey–Willam failure surface.

The given non-linear material behavior is depicted in Figure 5 by the equivalent stress $\sigma_c^{ef}$ and strain $\varepsilon_{eq}$, which is the product of equivalent stress and elastic modulus $E_{c,i}$ in $i$ direction. In the aforementioned Figure 5, the path of loading progresses up to point $D$, after which unloading starts. It is visible that the stress–strain relationship is not unique; rather, depended on the previous steps. The unloading starts with the change of equivalent stress increment sign. Particularly, if the loading starts after unloading was finished, the unloading direction is formed to the last point of loading point $D$.

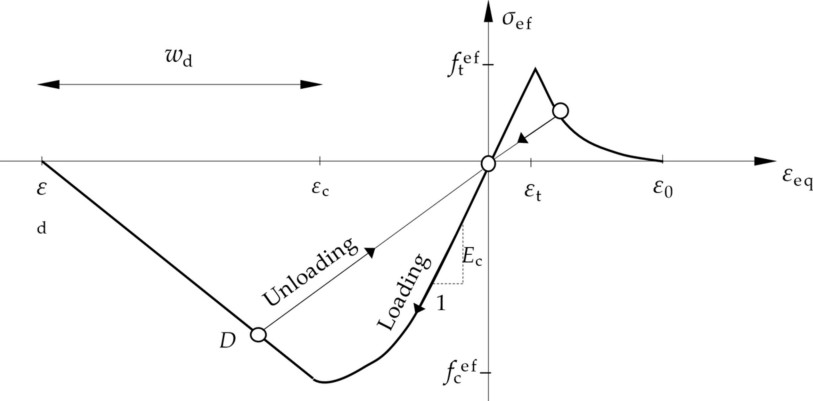

**Figure 5.** Standardized stress–strain relationship.

Only the compressive strength ($f_c$) of the concrete was tested, while the tensile ($f_t$) was calculated. Therefore, the tensile strength was recalculated using Equation (1) [18].

$$F_t = 0.44\sqrt{f_c} \qquad (1)$$

In compression, the endpoint of the softening part of the curve is characterized by the plastic displacement $w_d$ (Figure 5). By controlling the plastic displacement, the energy of a unit area of the failure surface is indirectly defined. From the experiments by Van Mier (1984) [19], the value of plastic displacement for regular concrete is $w_d$ = 0.5 mm. In this study, the plastic displacement was varied in both masonry and concrete material with $w_d \in \{-0.1, \ldots, -0.5\}$ mm.

The biaxial failure criterion was adopted from the works of Kupfer and Gerstle (1973) [20], as shown in Figure 6. The index numbers 1 and 2 present the principal stresses, while $f_c$ is the compressional strength of concrete cylinder that is predicted under the assumption that the path of stress is proportional under biaxial stress. In the tension-compression state, the failure function continues linearly from point $\sigma_{c,1}$ = 0 and $\sigma_{c,2} = f_c$ into the tension-compression region with linearly decreasing strength.

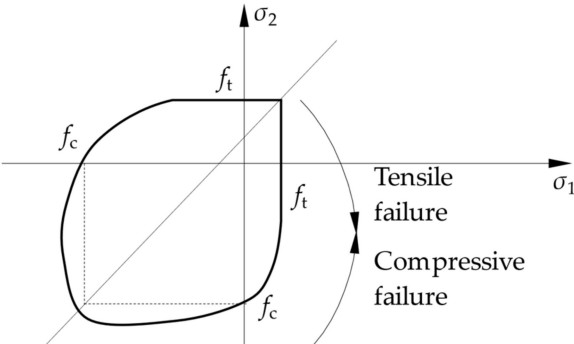

**Figure 6.** The standardized biaxial failure function.

The direction of plastic flow $\beta$ in the Drucker–Prager Plasticity Model is described by the return mapping algorithm for the plastic model that is based on the predictor–corrector approach as shown in Figure 7. During the corrector phase of the algorithm in Figure 7, the failure surface moves along the horizontal axis to simulate the hardening and softening of concrete. Concisely, the negative signs of plastic flow cause volume to shrink, positive to expand and 0 to continue unaffected. The negative values are recommended by Cervenka et al. (1998) [16] to decrease material volume if there is crushing. The direction of plastic flow was varied from negative to positive values for both the concrete $\beta \in \{-0.5, -0.4, \ldots, 0.2, 0.5, 1.0\}$ and masonry material model $\beta \in \{-0.5, -0.1, -0.05, 0\}$.

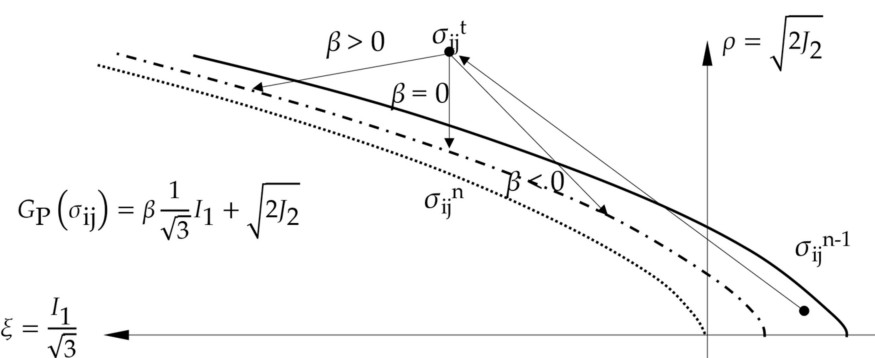

**Figure 7.** Plastic predictor-corrector algorithm.

Due to the presence of reinforcement in the concrete, cracks cannot fully develop. Hence, the concrete ends up contributing to steels stiffness, so-called tension stiffening. The coefficient that regulates the effects is denoted as $c_{ts}$, and it limits the tensile stress so it can not fall under the product of tensile strength and the coefficient $f_t c_{ts}$ (Figure 8). The recommended default value is $c_{ts} = 0.4$, and it was left untouched with the exception of one model tested without softening ($c_{ts} = 0$).

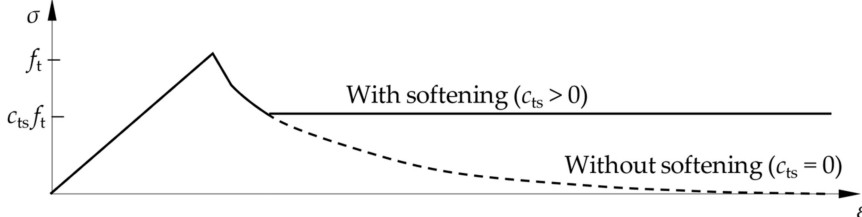

**Figure 8.** Tension stiffening.

Tensional fracture energy $G_F$ determines the material's resistance to crack propagation [15] as it can, for example, modify the line failure. The fracture energy presents the area under the tensile stress–displacement curve (Figure 9). Hence, if not tested experimentally, as in this case, an empirical calculation could be used based upon concretes mechanical properties. The software developers recommend using Equation (2) [16]. Other approaches were considered trough Equations (3)–(7), provided by Fédération internationale du béton (2013) [21].

$$G_F = 0.0000025\, f_t \tag{2}$$

$$G_F = G_{F0} \left( \frac{f_{cm}}{f_{cmo}} \right) \tag{3}$$

$$G_F = G_{F0} \ln \left( 1 + \frac{f_{cm}}{f_{cmo}} \right) \tag{4}$$

$$G_F = G_{F0} \ln \left( 1 - 0.77 \frac{f_{cm}}{f_{cmo}} \right) \tag{5}$$

$$G_F = G_{F0} \left( \frac{f_{cm}}{f_{cmo}} \right)^{0.18} \tag{6}$$

$$G_F = 73 f_{cm}^{0.18} \tag{7}$$

where $G_{F0} = 0{:}03$ MPa is fracture energy based on max aggregate size of 16 mm and $f_{cmo} = 10$ MPa [21].

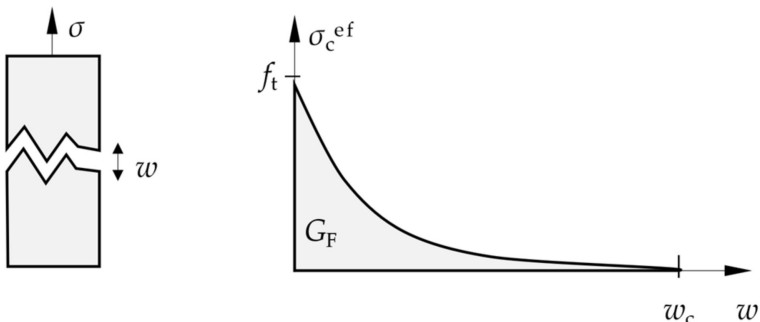

**Figure 9.** Fracture energy.

The effect of compressional strength reduction after the concrete had cracked is regulated within the compression field theory [22]. Within the computational software, the reduction factor $k_{red}$ can be modified by the user; therefore, the code developers have arranged function (Figure 10) from several experiments to accommodate user input [16]. From Figure 10, it is visible that for the zero normal strain, $\varepsilon_1 = 0$ here is no strength reduction $k_{red} = 1$, and in the case of large strains, the strength asymptotically approaches the minimum value $f_c^{ef} \simeq k_{red} f_c$.

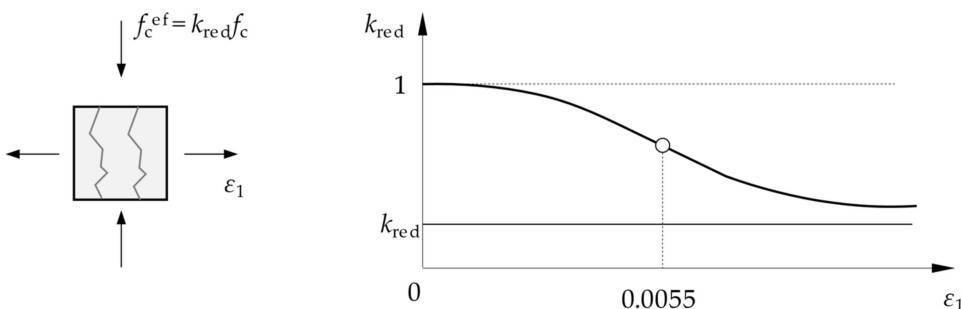

**Figure 10.** Compressive strength reduction of cracked concrete.

The value of the reduction has been determined as $k_{red} = 0.4$ by Kollegger and Mehlhorn (1988) [23]; however, Dyngeland (1989) [24] stated reduction as $k_{red} \geq 0.8$, which is also recommended by Cervenka et al. (2012) [16]. Values of $k_{red} \in \{0.8, 0.7, \ldots, 0.4\}$ were tested for masonry, and values of $k_{red} \in \{0.8, 0.7\}$ for concrete material model.

Range of crack models are available for selection in the computational software: Fixed (FCMC = 1), Rotated crack model (FCMC = 0), and everything in between (FCMC $\in \langle 0,1 \rangle$).

The fixed crack model has its crack direction in line with the principal stress direction (Figure 11b) at the moment of crack initiation. Additionally, due to the assumption of isotropy, stress and strain directions coincide in uncracked concrete. When loading further, the directions are, nevertheless, fixed and represent the material axis of orthotropy. Thus, orthotropy is introduced after cracking. Whereas the weak material axis $m_1$ is normal and strong $m_2$ is parallel with the cracks (Figure 11b). Since principal strain axis $\varepsilon_1$ and $\varepsilon_2$ can rotate and not coincide with the axis of orthotropy, results in additional shear stress on the crack face (Figure 11a).

The rotated crack model has the direction of principal strain in line with the principal stress axis. Thus, no shear stress is formulated on the crack plane, only two normal stress components. If the principal strain axes rotate during the loading, then the direction of the cracks would rotate as well.

Only fixed and rotated crack models were tested on concrete and masonry material models.

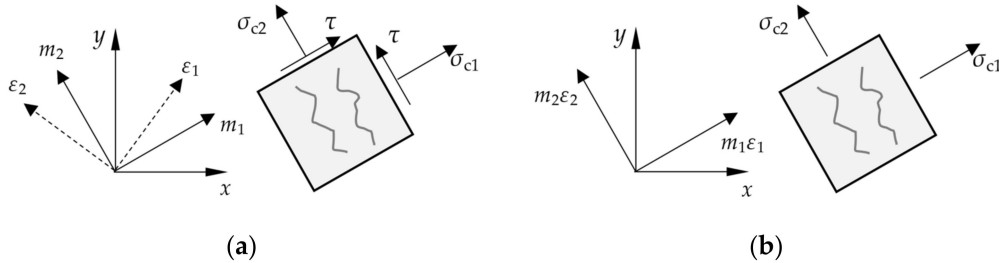

**Figure 11.** Crack models. (**a**) Fixed; (**b**) rotated.

In overall, the properties with which the simulations were initially started are presented in Table 2. They were adopted from their 2D micro-model counterpart [17].

**Table 2.** Fracture-plastic constitutive model material model initial properties.

| Description | | Frame Concrete | Concrete Lintel | Clay Block | Unit |
|---|---|---|---|---|---|
| Elastic modulus | $E$ | $4.100 \times 10^4$ | $3.032 \times 10^4$ | $5.650 \times 10^3$ | MPa |
| Poisson's ratio | $\mu$ | 0.200 | 0.200 | 0.100 | / |
| Tensile strength | $f_t$ | 4.000 | 2.317 | 0.380 | MPa |
| Compressive strength | $f_c$ | $-5.800 \times 10^1$ | $-2.550 \times 10^1$ | $-1.750 \times 10^1$ | MPa |
| Specific fracture energy Equation (3) | $G_f$ | $1.200 \times 10^{-4}$ | $5.739 \times 10^{-5}$ | 0.450 | N/mm |
| Crack spacing | $s_{max}$ | 0.125 | 0.125 | / | m |
| Tensile stiffening | $c_{ts}$ | 0.400 | 0.400 | / | / |
| Critical compressive disp. | $w_d$ | $-5.000 \times 10^{-4}$ | $-5.000 \times 10^{-4}$ | $-5.000 \times 10^{-4}$ | / |
| Plastic strain at $f_c$ | $\varepsilon_{cp}$ | $-1.417 \times 10^{-3}$ | $-8.411 \times 10^{-4}$ | $-1.358 \times 10^{-3}$ | / |
| Reduction of $f_c$ due to cracks | $k_{red}$ | 0.800 | 0.800 | 0.800 | / |
| Crack shear stiffness factor | $S_F$ | $2.000 \times 10^1$ | $2.000 \times 10^1$ | $2.000 \times 10^1$ | / |
| Aggregate size | | $1.600 \times 10^{-2}$ | $2.000 \times 10^{-2}$ | / | m |
| Fixed crack model coefficient | | 1.000 | 1.000 | 1.000 | / |

### 2.4.2. Interface Material Model

The zero-thickness interface model is used to simulate contact between two solid macroelements; hence, concrete–masonry and masonry–masonry contact. The interface material is based on the Mohr–Coulomb criterion without tension.

$$\tau \begin{cases} \leq c - \sigma\phi, & \sigma \leq 0 \\ = \tau_0 \sqrt{1 - \dfrac{(\sigma - \sigma_c)^2}{(f_t - \sigma_c)^2}}, & 0 < \sigma < f_t \\ = 0, & \sigma < \tau \end{cases} \tag{8}$$

Briefly, the initial failure surface (Figure 12) corresponds to the *Mohr–Coulomb condition* with ellipsoid in tension regime. After the stresses violate condition under Equation (8), the surface collapses to a residual one, which corresponds to *dry friction* $\sigma\phi$. The ellipsoid is formed by two tangents, and the vertical one is perpendicular to the normal stress axis $\sigma$ at tensile strength $f_t$.

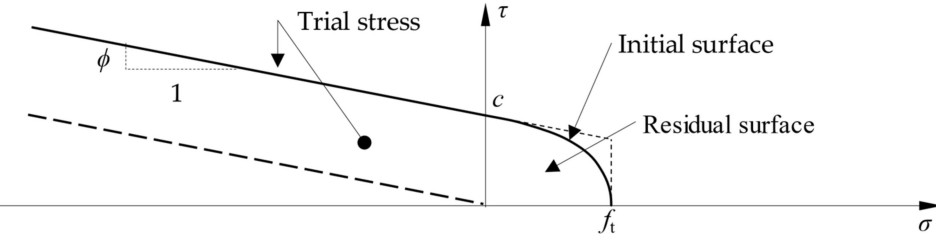

**Figure 12.** Failure surface for interface elements.

The interface model includes both *tangential* $K_{tt}$ and normal stiffness $K_{nn}$. Cervenka et al. (2012) [16] suggest using Equation (9), while Equation (10) was suggested in Diana fea bv (2019) [25]. Additionally, there are two stiffness counterparts $K_{nn}^{min}$ and $K_{tt}^{min}$. Their assignment is purely numerical, i.e., to avoid infinite global stiffness after the interface fails and its stiffness reaches 0. Recommend values are a thousand parts of $K_{nn}$ and $K_{tt}$ [9].

$$K_{nn} = \frac{\min\{E_i\}}{t}, \; K_{tt} = \frac{\min\{G_i\}}{t}, \tag{9}$$

where $\min\{E_i\}$ is the minimal elastic modulus of the material surrounding the interface element and $t$ is the thickness of the contact element.

$$K_{nn} = \frac{E_U E_J}{t_j(E_U - E_J)}, \; K_{tt} = \frac{G_U G_J}{t_J(G_U - G_J)}, \tag{10}$$

where $_U$ is the greater value of moduli surrounding the contact element.

Within the software, it is possible to define multi-linear evolution laws for tensile and shear softening. With those laws, it is possible to simulate degradation of tensile strength caused by shear stress and vice versa (Figure 13). For example, if there is no softening law implemented within the tensile behavior, stress drops to zero after reaching tensile strength (Figure 13b). Likewise, in shear behavior, stress drops to the value of *dry friction* $\sigma\phi$. However, if the softening law is introduced, the behavior resorts to softer and gradual degradation (dashed lines in Figure 13a,b).

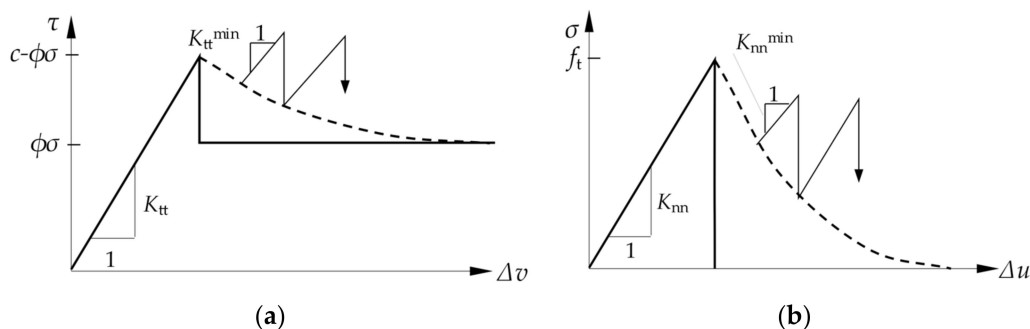

**Figure 13.** The behavior of the interface model. (**a**) In shear; (**b**) in tension.

The softening laws were implemented to simulate *mortar interlocking* described by Penava et al. (2016) [17]. The mortar interlocking is an effect that occurs in hollow masonry units that were bonded by mortar. When the mortar is laid on top of the blocks, it falls into its voids adding to mortar below, but encased in the voids. Thus, effectively producing more shear resistant and monolith connection, that causes *tensional* rather than *sliding failure* when testing masonry triplets. Hence, the shear interface function has had both hardening and softening parts (Equation (11) and Figure 14a), while the tensile function only the softening part (Figure 14b). The shear function was calculated using Equation (11), where the endpoint of the softening curve ($v = 2$ mm) was set by trial and error.

$$c = \begin{cases} c_0, & v = 0.00 \text{ mm} \\ 0.065\,f_{mu}, & v = 0.04 \text{ mm} \\ 0, & v = 2.00 \text{ mm} \end{cases} \tag{11}$$

In overall, the properties with which the simulations were initially started are presented in Table 3. They were adopted from their 2D micro-model counterpart [10].

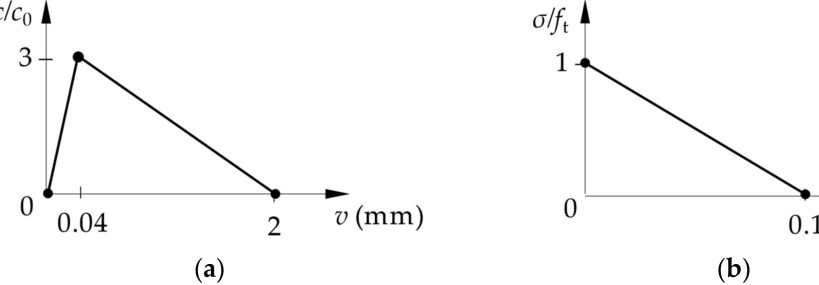

**Figure 14.** Multi-linear interface functions (initial values). (**a**) Cohesion softening/hardening function; (**b**) tensile softening function.

**Table 3.** Initial interface material properties.

| Description | Symbol | Bedjoint | Headjoint | Unit |
|---|---|---|---|---|
| Normal stiffness (Equation (8)) | $K_{nn}$ | $5.65 \times 10^5$ | $8.50 \times 10^4$ | MPa |
| Tangential (shear) stiffness (Equation (8)) | $K_{tt}$ | $2.57 \times 10^5$ | $3.86 \times 10^4$ | MPa |
| Tensile strength | $f_t$ | 0.20 | 0.20 | MPa |
| Cohesion | $c$ | 0.35 | 0.35 | MPa |
| Friction coefficient | $\phi$ | 0.24 | 0.24 | / |
| Interlocking function | Figure 14 | Where applicable * | / | |

\* Only on interfaces between masonry units.

### 2.4.3. Other Material Models

Discrete reinforcement was modelled by 1D truss elements. The bilinear law with hardening was used and simultaneously modified by the Menegotto and Pinto nonlinear model. The recommended values were used, only user input was that of yielding $\sigma_y$ and tensile strength $\sigma_t$. The rebar material model values are presented in Table 4.

**Table 4.** Bilinear steel material properties.

| Description | Symbol | Value | Unit |
|---|---|---|---|
| Elastic modulus | $E$ | $2.10 \times 10^5$ | MPa |
| Yield strength | $\sigma_y$ | $5.50 \times 10^2$ | MPa |
| Tensile strength | $\sigma_t$ | $6.50 \times 10^2$ | MPa |
| Limited ductility of steel | $\varepsilon_{lim}$ | 0.01 | / |
| Bauschinger effect exponent | $R$ | 20.00 | |
| Menegotto-Pinto 1st parameter | $C_1$ | 0.92 | |
| Menegotto-Pinto 2nd parameter | $C_2$ | 0.12 | |

Elastic plates used a simple *homogeneous ideal elastic-plastic* material model with an elastic modulus of $E$ = 200,000 MPa and Poisson's ratio of $\mu$ = 0.3.

During the IP experiments, the gravity force of 365 kN was applied on top of the columns, where steel rollers were positioned. Steel rollers freed the translation; however, with such a high normal load an opposing *friction force* occurs in the direction opposite to the translation. The friction of steel rollers is small, about $\phi$ = 0.03 [26]. However, when accounted for the gravity load, friction force $T_F$ adds to approx. 10 kN (Equation (12)) for one column. The effects of added friction were modelled as *multi-linear spring*, where $K_s$ stiffness is the result of divided 2 friction forces (2 columns) and beam's area where the spring was placed (Eq. 13). The spring was modelled with a small *hardening* part (0.83 → 0.85) to stabilize the computations (Figure 15).

$$T_F = 365 \, \phi \approx 10 \text{ kN} \tag{12}$$

$$K_S = 2T_F / A_{Beam} = 0.85 \text{ MPa} \tag{13}$$

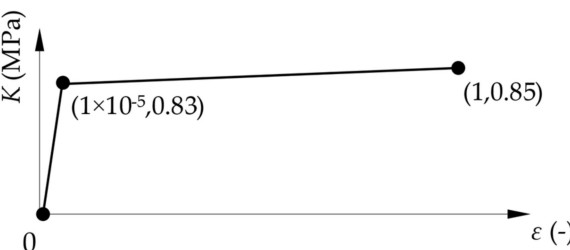

**Figure 15.** Multi-linear spring.

The overview of parameters changed can be seen in Table 5. In the column: model refers to the material model; property to the parameter that was varied; reference provides citations on which the change was based on; range/value to the range or the value of the parameter (property) that was varied; value old/new the inherent or adopted values from the range/value column. The table does not specify what part of the test simulation (IP or OoP) prompted the adaption of a certain parameter, which is described in Section 3. Results and discussions.

**Table 5.** Varied parameters.

| Model | # | Property | | Reference | Range/Value | Value Old | Value New | Unit |
|---|---|---|---|---|---|---|---|---|
| Concrete | 1 | FCMC * | | | 0.00, 1.00 | 1.00 | 1.00 | / |
| | 2 | $w_d$ | | [19,27] | −0.10, −0.20, … , −0.50 | −0.10 | −0.10 | mm |
| | 3 | $f_t$ | Equation (1) | [18] | 3.35 | 4.00 | 3.35 | MPa |
| | 4 | $k_{red}$ | | [16,22] | 0.70 | 0.80 | 0.80 | / |
| | 5 | $\beta$ | | [28] | −0.5, −0.4, … , 0, 0.1, 0.2, 0.5, 1 | 0.00 | −0.10 | / |
| | 6 | $G_f$ | Equation (2) Equation (3) Equation (4) Equation (5) Equation (6) Equation (7) | [16] [21] | 0.084 0.103 0.125 0.156 0.151 152,000.000 | 0.084 | 0.084 | N/mm |
| Masonry | 1 | $w_d$ | | [19,27] | 5.00, 0.50, 0.05, 0.005 | 0.50 | 0.50 | mm |
| | 2 | $f_{c,0} = f_c$ | | | −58.00 | −5.14 | −5.14 | MPa |
| | 3 | FCMC * | | | 0.00, 1.00 | 1.00 | 0.00 | / |
| | 4 | $k_{red}$ | | [16,22] | 0.8, 0.7, 0.6, 0.50, 0.49, … , 0.4 | 0.80 | 0.40 | / |
| | 5 | $\varepsilon_{cp}$ | | | −1.0, −0.1 | −1.36 | −1.36 | ‰ |
| | 6 | $\beta$ | | [16] | 0, −0.05, −0.10, −0.20, −0.30 | 0.00 | 0.00 | / |
| | 7 | $f_t$ | | [18] | 0.38 | 1.80 | 0.38 | MPa |
| Interface | 1 | max($c/c_0$) $x$ | | | 100, 10, 1, 0.10, 0.25, 0.50, 0.75 | 2.00 | 0.75 | mm |
| | 2 | max($c/c_0$) | | | 1.0, 1.5, 2.0, 2.5, … , 5.0 | 3.00 | 3.00 | / |
| | 3 | $K_N$ (Equation (10)) | | [25] | 140, 140, 0.140 | 565.00 | 565.00 | N/mm³ |
| | | $K_T$ (Equation (10)) | | | 14, 1.4, 0.014 | 275.00 | 275.00 | N/mm³ |
| | 4 | max($\sigma/f_t$) | | [14] | 3.0, 2.5, 2.0, … , 0.5 | 1.00 | 1.00 | / |
| | 5 | max($\sigma/f_t$) $x$ | | | 0.01 | 0.10 | 0.01 | mm |

Additionally, numerous models were computed with combined parameters to investigate their interaction (e.g., $r_{c,lim}$ with and without $\beta$; cohesion and tension softening parameters, etc.). * FCMC—fixed crack model coefficient. max{$f(x)$} $x$—Maximal $x$ values of a given function $f(x)$.

## 2.5. Boundary Conditions

The boundary conditions were set to mimic the experimental ones. The ones that represent IP and OoP tests on frames with and without infill wall and openings is presented in Figure 16. Similarly, the boundary conditions of the masonry wall tested on OoP bending is presented in Figure 17.

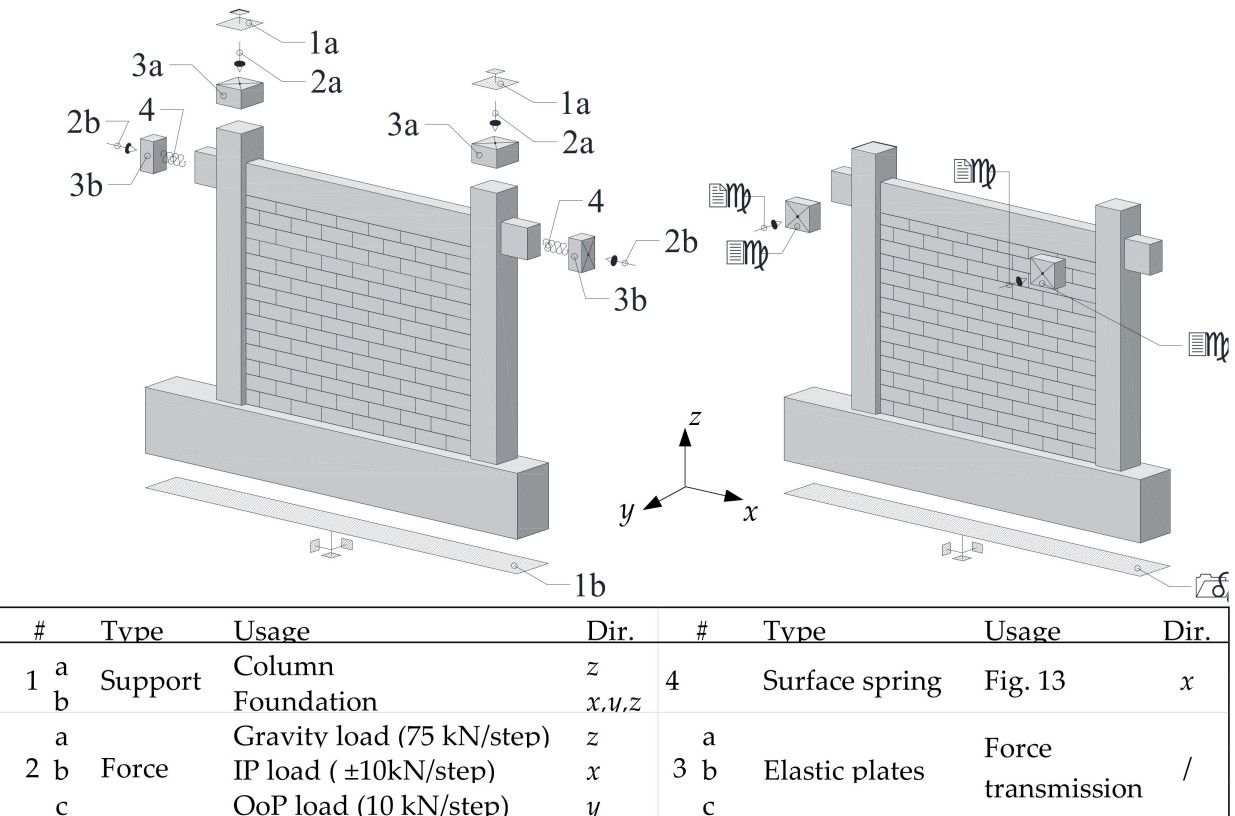

| # | | Type | Usage | Dir. | # | Type | | Usage | Dir. |
|---|---|------|-------|------|---|------|---|-------|------|
| 1 | a | Support | Column | z | 4 | Surface spring | | Fig. 13 | x |
| | b | | Foundation | x,y,z | | | | | |
| 2 | a | Force | Gravity load (75 kN/step) | z | 3 | a | Elastic plates | Force transmission | / |
| | b | | IP load ( ±10kN/step) | x | | b | | | |
| | c | | OoP load (10 kN/step) | y | | c | | | |

**Figure 16.** Boundary conditions of FI model ((**left**) IP, (**right**) OoP loadings).

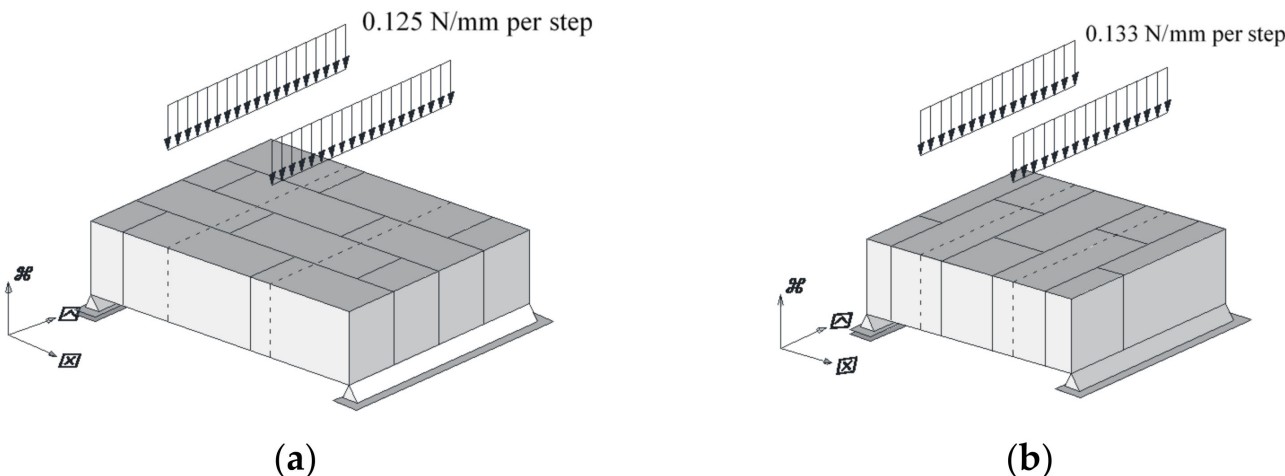

**Figure 17.** Boundary conditions for OoP masonry wall bend tests. (**a**) Perpendicular to bedjoints; (**b**) parallel with bedjoints.

In the IP micromodel simulations, the gravity force of 365 kN was introduced in 5 steps per 75 kNs. The gravity load was followed by the IP forces, introduced with ±10 kN/step following the loading protocol (Figure 18a). The protocol had a double peak force repetition. However, unlike the experimental ones, the protocol for computational models did not have the *pushover* part (Figure 18). The double force repetitions were used also for OoP cyclic, quasi-static simulations; however, it was administrated unidirectionally (Figure 18b). Note that in the case of OoP drift driven tests, gravity load was not implemented.

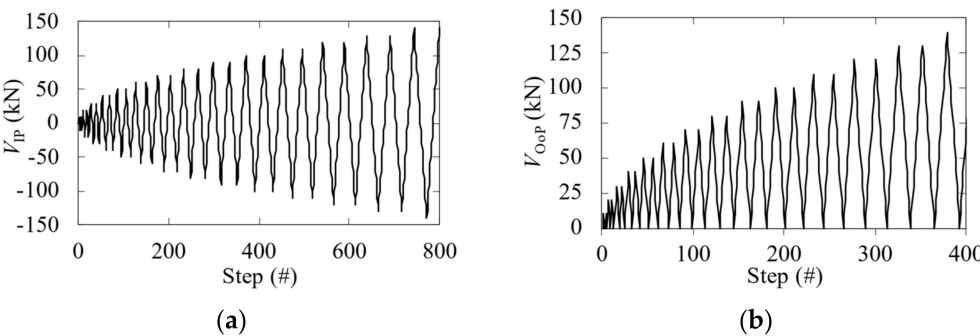

**Figure 18.** Cyclic, quasi-static load protocols. (**a**) IP; (**b**) OoP drift force protocol.

The OoP bend test computational models of masonry walls had line loads, that when multiplied by their length equal to 5 kN/step. The loading protocol for these models was purely monotonic. Note that dashed lines in Figure 17 present a perfect connection. It was created so that the load may transfer via its upper line. Additionally, the dotted line presents the cropped pieces of the model, as they were not necessary for the simulations.

## 3. Results and Discussion

### 3.1. Cyclic, Quasi-Static IP Loading on Masonry Infilled RC Frames

Initially, the 3D had a greater response than its 2D micromodel counterpart or the experimental response in the case of IP loading (Figures 19 and 20). Hence, further calibrations and sensitivity analyses were undertaken. Some parameters had no effect, while others did, either on the load-bearing or to the computational stability part (e.g., crack model, fracture energy). The varied parameters are presented in Table 5; whereas, the column *range/value* presents the variations of the parameter, *old* the inherent and *new* the accepted values. Note that not all values in the table were changed due to the IP test, some were caused by OoP tests. The parameters that were adopted purely for IP calibrations were: (1) Crack model of infill unit (*fixed* → *rotated*); (2) concrete's direction of plastic flow ($\beta$: $0.0 \rightarrow -0.1$); (3) masonry's reduction of compressive strength due to cracks ($k_{red}$: $0.8 \rightarrow 0.4$); (4) concrete's tensile strength ($f_t$: $4.00 \rightarrow 3.35$ MPa). With the newly adopted parameters, the models were considered calibrated in the IP direction.

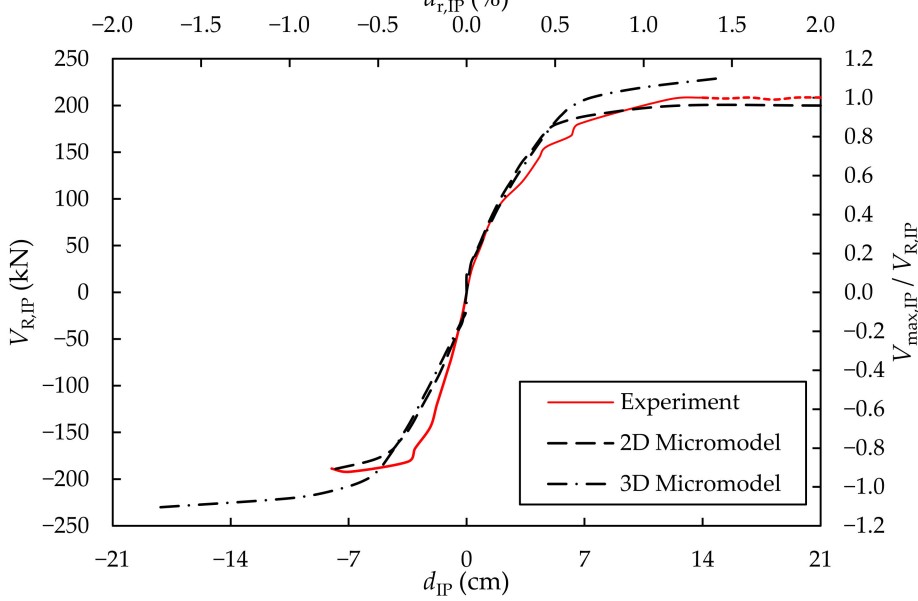

**Figure 19.** 2D vs. 3D micromodel approach on BF model.

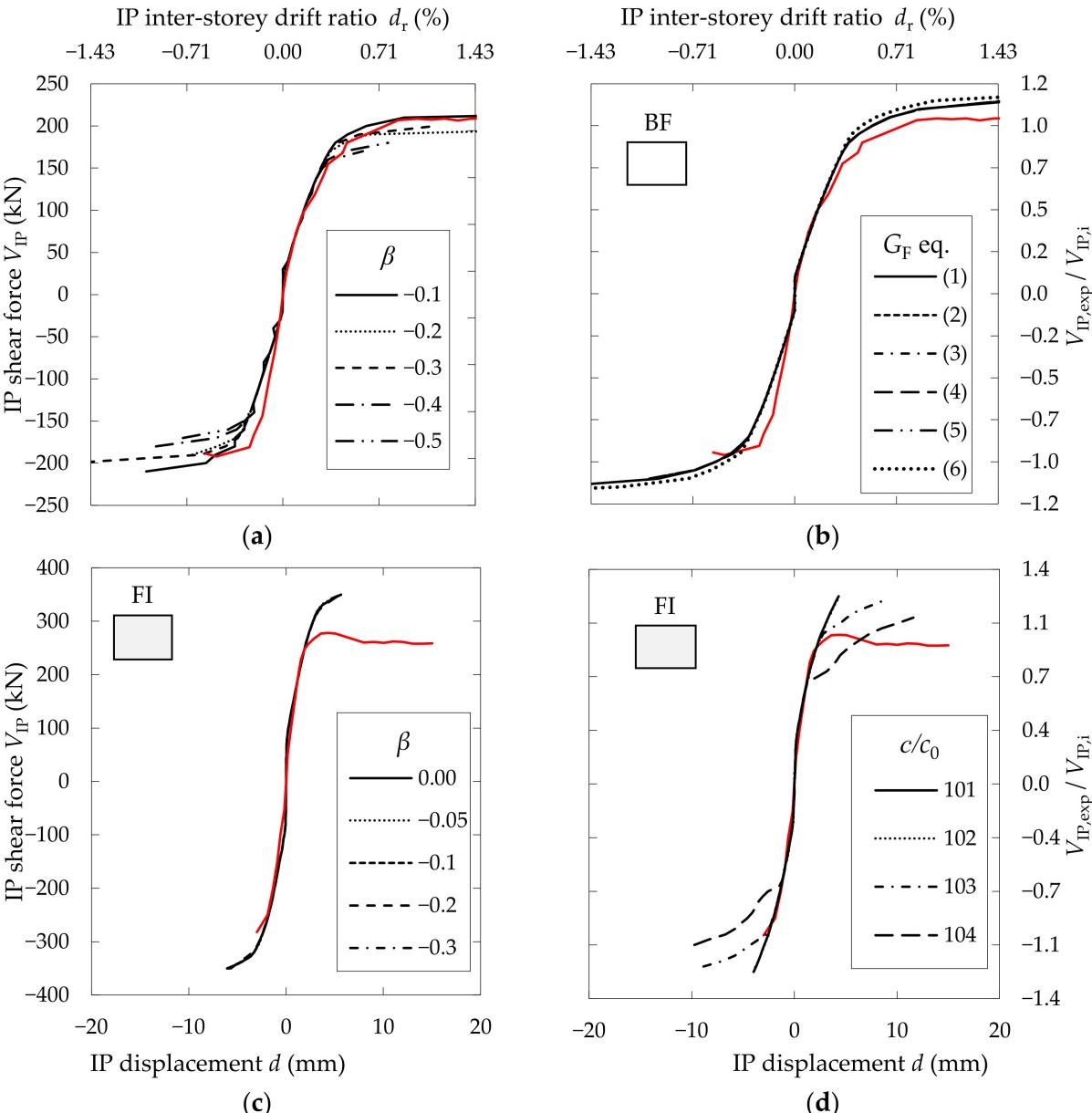

**Figure 20.** Examples of IP calibration process: (**a**) bare rc frame case $\beta$ parameter influence investigation; (**b**) bare rc frame case $G_F$ parameter influence investigation; (**c**) full-infilled rc frame case $\beta$ parameter influence investigation; (**d**) full-infilled rc frame case $c/c_0$ parameter influence investigation.

Due to a large number of changes to the material model parameters (Table 5), few of them were selected and displayed in Figure 20. From the analysis of the results, it was observed that some parameters had more effects than others. Yet, some did not show a noticeable difference in the models' response, as some impacted computational stability. For instance, the change of the masonry's crack model from fixed to rotated was done purely to ensure stable computations in case of models with openings, while other parameters lower the computational response.

Interestingly, a significant shift in the model's behavior caused by a change in concrete did not mirror the same nor similar effects when it came to the masonry material model (Figure 20). This is best illustrated in the significantly lower model response caused by reducing the tensile strength of concrete by 18%. Yet, no significant difference was observed when masonry's tensional strength was reduced by 80%. Likewise, in the case of different parameters as the direction of plastic flow, etc. From the results, it was observed that within

the micromodels infill wall, the changes were more prominent when interface material model properties were tampered with (Figure 20—lower right graph), not the masonry.

The computational against experimental crack patterns are presented in Figure 21. It is visible that they developed very similar patterns and failure mods, i.e., crack patterns that resemble compression strut and sequenced failures, bedjoint sliding. In more detail, the models followed *typical* failure modes: (1) BF model developed plastic hinges and ended with flexural failure (Figure 21a); (2) FI model had bedjoint sliding failure (Figure 21b); (3) CD had a diagonal tensional failure of masonry piers developed on the sides of the opening (Figure 21c); (4) CW had a diagonal tensional failure of masonry piers developed on the sides of the opening (Figure 21d); (5) ED had a diagonal tensional failure of masonry pier that developed on the side of the opening (Figure 21e); (6) EW bedjoint sliding failure of the masonry pier that developed on the side of the opening (Figure 21f).

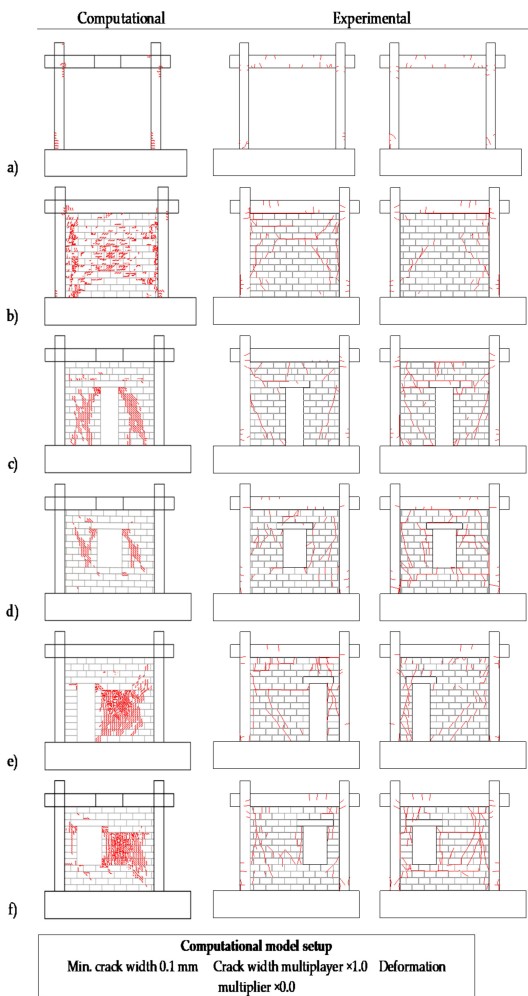

**Figure 21.** Crack patterns of IP experimental and computational models (results from models calibrated in both IP and OoP direction). (**a**) bare rc frame case; (**b**) full-infilled frame case; (**c**) centric door case; (**d**) centric window case; (**e**) eccentric door case; (**f**) eccentric window case.

The envelopes from calibrated numerical and experimental models are plotted in Figure 22. Whereas the pushover parts in the experiments were dashed, which the numerical model did not contain. As observed, the computational models were in good correlation with the physical ones; in terms of initial stiffness, yielding point, lateral resistance and ductility. It was measured that the micromodel was able to capture effects of load position in case of models with openings, especially those with eccentric ones. Namely, the part of the envelope that was realized by the force on the opposite side, and/or the side with

shorter wall adjacent to the opening had lower response. Therefore, when plotted against each other, the parts of the envelopes do not match up to the point when the infill wall loses its effect.

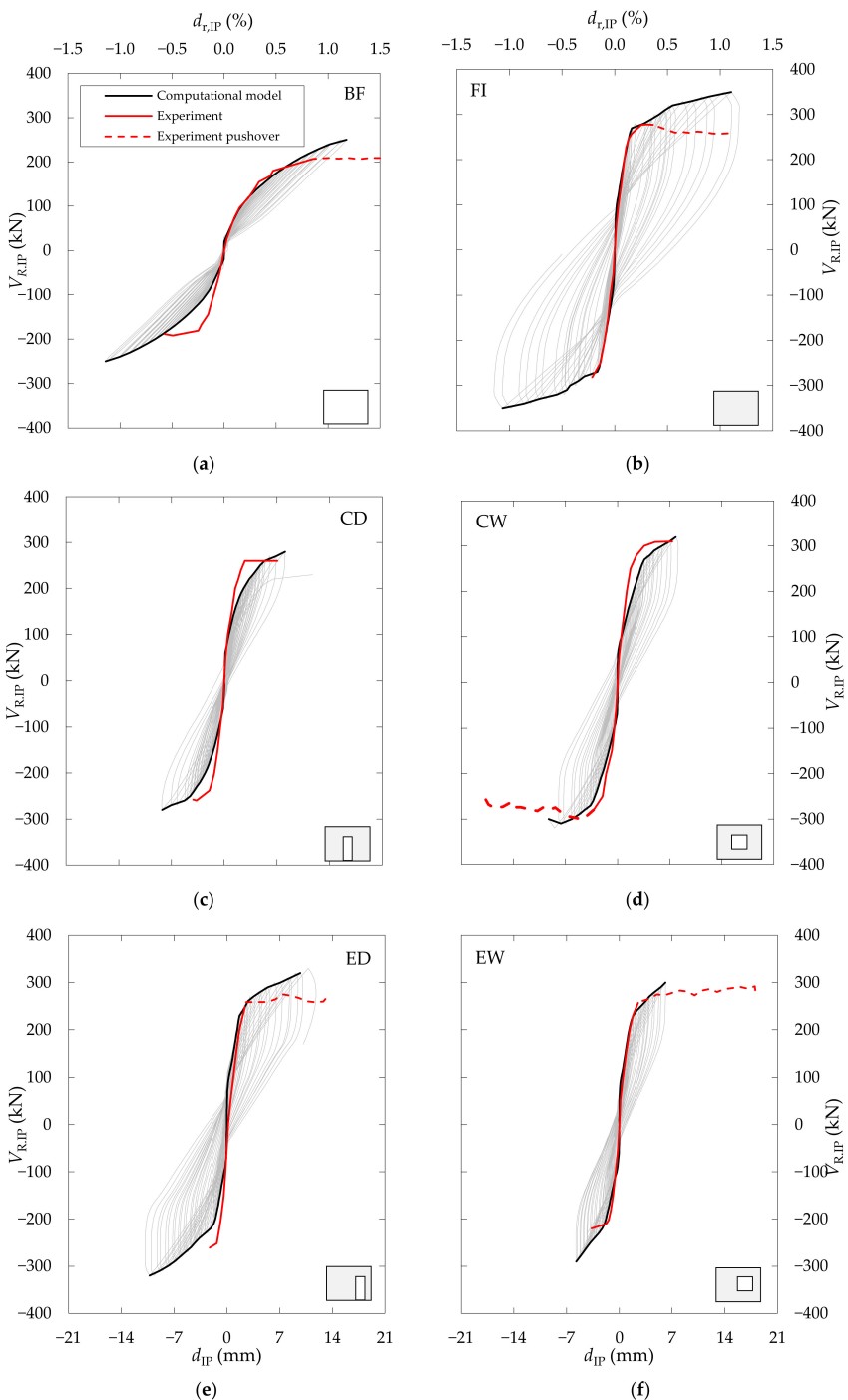

**Figure 22.** Calibrated cyclic, quasi-static IP micromodel response (results from models calibrated in both IP and OoP direction): (**a**) bare rc frame case; (**b**) full-infilled frame case; (**c**) centric door opening case; (**d**) centric window opening case; (**e**) eccentric door opening case; (**f**) eccentric window opening case.

In order to numerically evaluate the behavior of the models, the mean average percentage error (*MAPE*) was used (Eq. 14) The values are presented in Table 6, where they were divided into the positive (+) and negative (−) parts of the graph. The reason is that in the experimental part, the positive and negative curves do not align (Figure 22). The

effects of why are somewhat described in the paragraph above, the rest could be added to expected imperfections of the laboratory conditions. The values of shear forces were linearly interpolated at every $d_r = \pm 0.25\%$. Note that only the cyclic parts were compared, the pushover part from the experiments was discarded. From the table, it is visible that computational models have a good correlation with the experiment, whereas some parts had better correspondence in the negative than in positive parts and vice versa.

**Table 6.** *MAPE* (Equation (14)) in % (results from models calibrated in both IP and OoP direction).

| Part | BF | FI | CD | CW | ED | EW |
|------|------|------|--------|--------|-------|-------|
| + | −0.37 | −5.86 | −14.88 | −12.60 | −5.93 | 7.93 |
| - | −21.77 | 9.03 | −15.39 | −10.30 | −4.1 | −2.66 |
| Avg. | 11.07 | 7.45 | 11.23 | 11.62 | 5.01 | 5.3 |

Note that Figures 21 and 22 and Table 6 show the results from fully calibrated (IP and OoP) models.

$$MAPE = \frac{1}{n}\sum_{i=1}^{n}\frac{V_{i,exp} - V_{i,num}}{V_{i,exp}} \tag{14}$$

### 3.2. OoP Bend Test of Masonry Wall

The OoP bending test of the masonry wall was firstly computed with the values of calibrated IP computational model (Section 3.1). As expected, the values of the computational model were much greater than the experimental ones. The reason was due to the tensile strength of the masonry, which was set in the direction of voids. It was then changed to its *perpendicular* value ($f_t$: $1.80 \rightarrow 0.38$ MPa), which prompted the change in fracture energy based on Equation (2). The following calculations were unstable. This was aided by changing the interface's softening endpoint ($0.10 \rightarrow 0.01$ mm), and by reverting fracture energy $G_F$ to its old value, i.e., in direction of voids.

The calibrated models had the same mode of failure (Figure 23a,b vs. Figure 2b) as the experimental specimens. Likewise, the values of load-bearing force and maximum principal stresses were in good relation (Table 7).

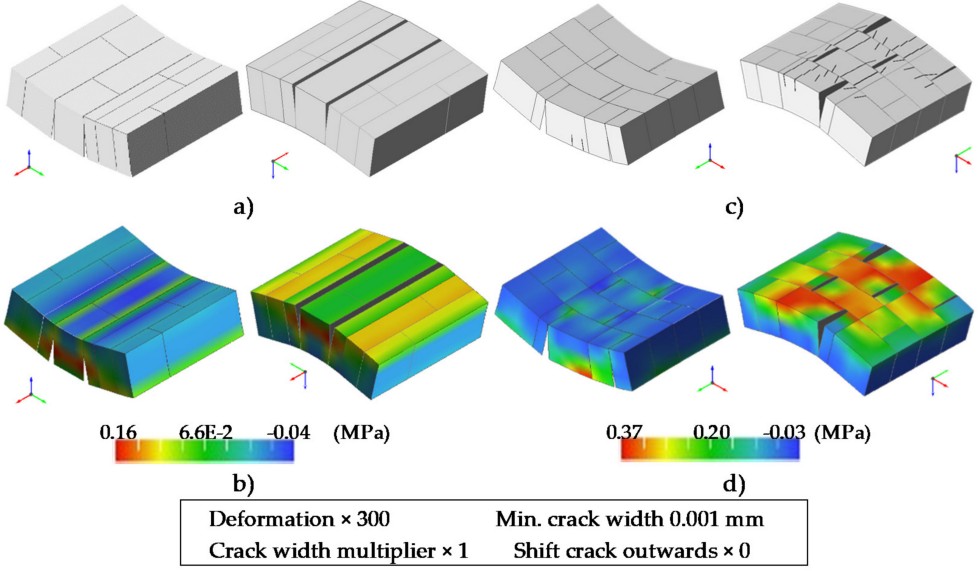

**Figure 23.** Crack and stress patterns on the calibrated computational models. (**a**) Deformed model; (**b**) Max. principal stress; (**c**) deformed model; (**d**) Max. principal stress.

**Table 7.** Comparison of calibrated computational and experimental models.

| Load in Relation to Voids | Computational/Experiment. Values | | Difference from the Experiment | |
|---|---|---|---|---|
| | $F_{max}$ (kN) | $\sigma_{max}$ (MPa) | $F_{max}$ (%) | $\sigma_{max}$ (%) |
| Parallel | 4.50/4.07 | / | 9.55 | / |
| Perpendicular | 6.20/6.69 | 0.37/0.38 | 7.32 | 2.63 |

Since there were crucial changes to the masonries and interface's parameters, the previously calibrated IP computational model certainty was questioned. The IP models were computed with the newly adopted parameters that produced insignificant changes and kept a good correlation with the IP experiments (Figures 21 and 22, Table 6).

*3.3. OoP Drift Driven Load on Frames with Masonry Infill Walls*

This part of the series was difficult to evaluate since the experimental frames had a history of previous damages. This is visible in the fact that the physical FI and CD specimens had the lowest load-bearing capacity (Figure 24b), lower even than the BF specimen. Therefore, the calibration or the evaluation of the computational models did not rely on the absolute values. Rather, it depended on the relative values and behavior. Furthermore, all physical and computational models had flexural fail mode. The flexural fail mode was also successfully simulated/calibrated in case of IP BF simulation.

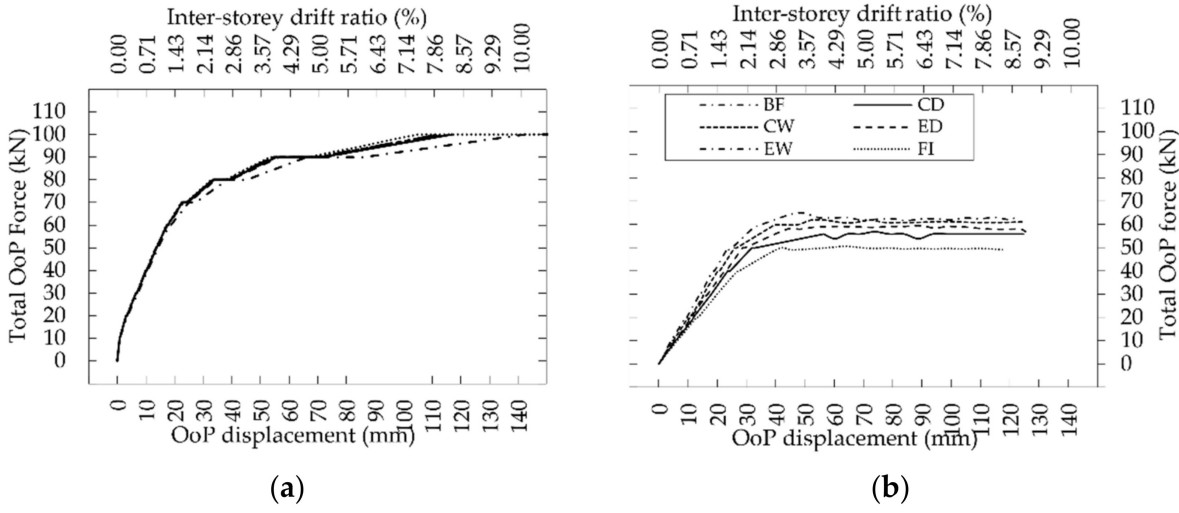

**Figure 24.** Load-capacity curves from OoP cyclic, quasi-static tests. (**a**) Computational model; (**b**) experimental specimens.

Both physical and computational specimens showed great stability, enduring to drift ratios of $d_r > 10\%$. Clearly, such drift ratios are unrealistic in real scenarios and inappropriate for engineering practices.

As expected, the computational model had a greater response in terms of load-bearing capacity and initial stiffness than its experimental counterpart (Figure 24). In Figure 24, the total force is the sum of forces from both columns. Furthermore, both approaches had yielding occurring at about 1.5% drift ratio, less energy dissipation, and differences between the models.

When comparing the damage states from Figure 25, it is visible that both approaches produced congruent results. The tension sides developed horizontal cracks along with the columns, and masonry infill wall (along bedjoints). The compression sides were observed crushing at the foot of columns and infill wall. In the case of openings, all developed cracking around the lintel. Frame suffered comparatively more damage than the infill wall. The computational models followed the same principle as experimental ones, where the frame–infill wall interaction was unidirectional, i.e., the frame transmitted displacements and with it damaged the infill wall. The opposite effects can be viewed as inconsiderable.

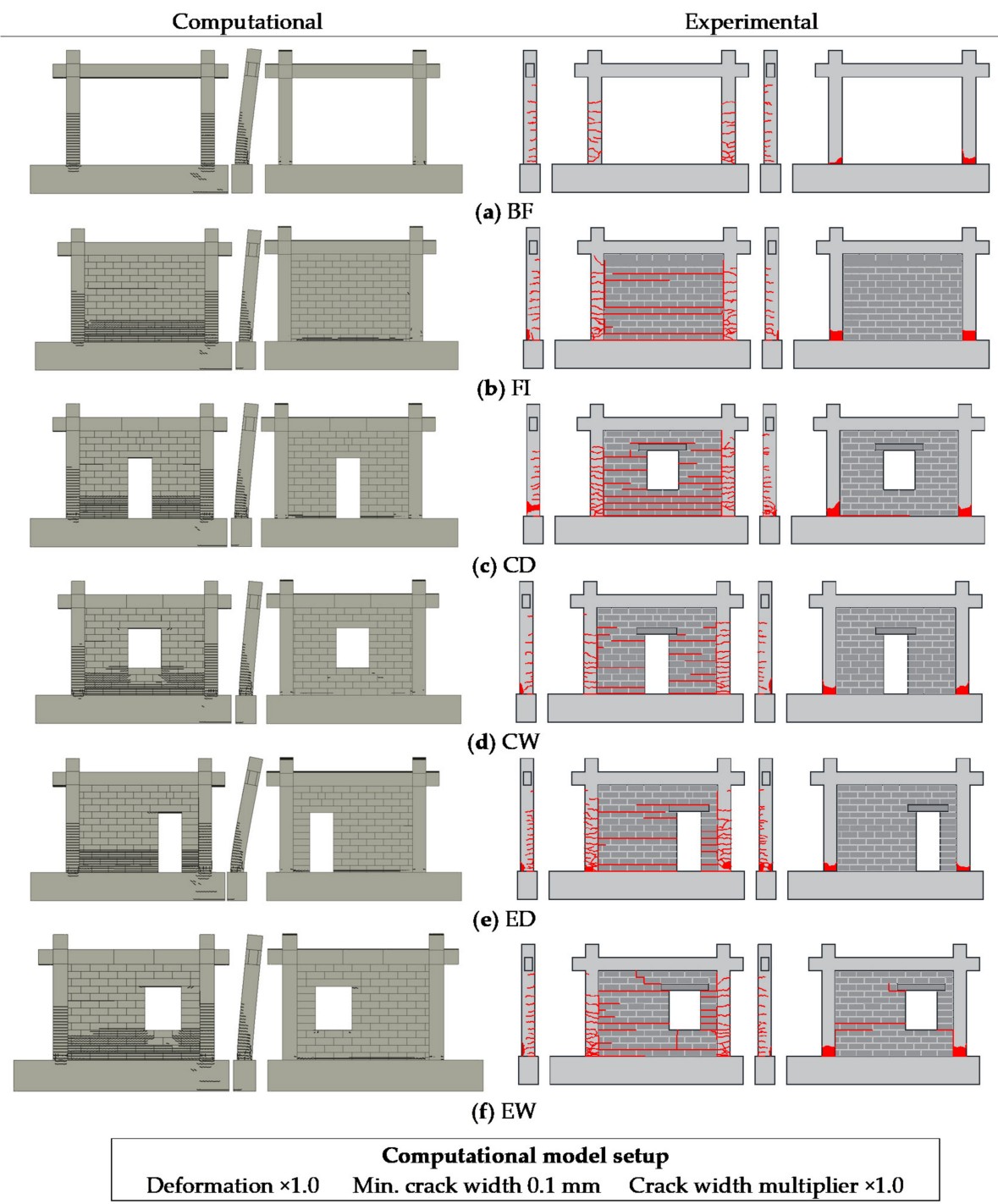

**Figure 25.** Crack patterns of OoP cyclic, quasi-static load (@ $d_r \cong 10\%$). (**a**) BF; (**b**) FI; (**c**) CD; (**d**) CW; (**e**) ED; (**f**) EW.

The computational models shed more insight into the behavior of the specimens that the experiments did not capture (Figure 24). It shows that there is an observable difference between the BF model and the rest. That is exaggerated after the point of yielding ($d_r \cong 1.4\%$). There is also a slight difference between the FI model and the rest; therefore, encasing the models with openings between the BF and FI model. If looked up close, then it is visible that models with the door had a lower response than those with window openings. However, those differences are meagre.

The displacement maps are presented in Figure 26, whereas the other models had similar patterns. From the displacement maps, it is visible that with both approaches the

frame and infill wall move as a single unit. Additionally, the displacement progressively raises from the foundation to the top of the columns.

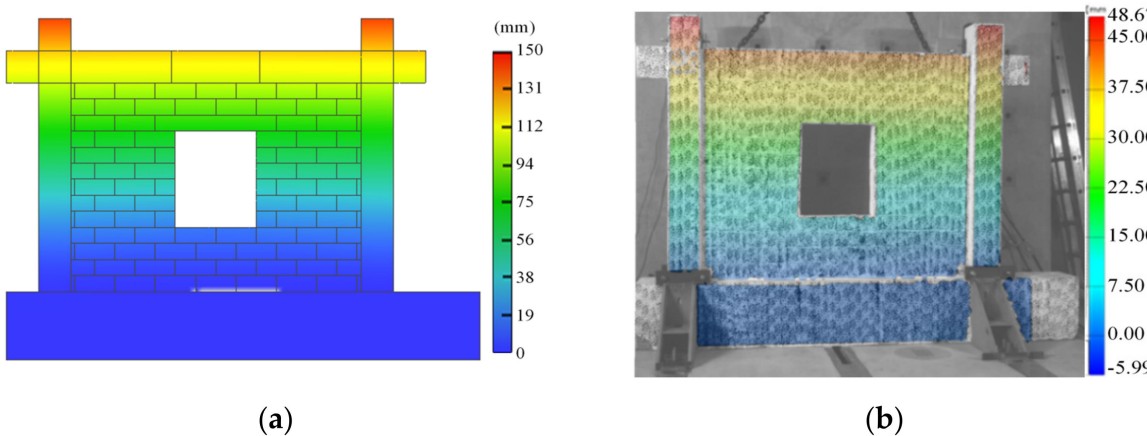

(**a**)                                         (**b**)

**Figure 26.** OoP displacement maps of CW model (others had nearly the same pattern). (**a**) Computational model (@ $d_r \cong 10\%$); (**b**) Experimental model (@ $d_r \cong 1.5\%$).

Similar findings to those listed above were found in single-story; -bay dynamical experiments. Where there was less energy dissipation, the infill wall and frame acted as one superstructure, frame failed rather than the infill [29,30]. Likewise, the same observations are in line with the two OoP drift-driven tests [31–33].

### 4. Summary and Conclusions

This paper presents the development and calibration process of 3D micromodels that simulated IP and OoP cyclic, quasi-static drift loads on frames with and without infill walls and openings, and also OoP bend loads on masonry walls. The purpose of micromodel development was to use those models in later studies to gain insight into the gaps that were found in the field research, e.g., combination of IP and OoP drift-driven loads with various openings.

The micromodels were developed based on the experimental test series: (1) IP cyclic-quasi static tests on frames with/without infill walls and openings [5]; (2) OoP bending test of masonry walls with load perpendicular and parallel to bedjoints [14]; (3) OoP drift driven, cyclic, quasi-static tests on frames with/without infill walls and openings [15]. All test series had the same geometrical and mechanical properties. The material models were initially adopted from a 2D, IP micromodel [17]; however, it yielded a stronger response (Figure 19). Therefore, a calibration was needed.

From the calibration process, it was found that some parameter variations resulted in a considerable change in behavior, while others did not. It was found that even within the same material model, changes to the concrete did not mirror the same effects on the model's behavior as the masonry model. For instance: (a) the direction of plastic flow had a great influence on the concrete and not on the masonry material model (Figure 23); (b) a small change ($\cong16\%$) in tensile strength had greater effects on concrete than larger change ($\cong80\%$) on the masonry material model; (c) concrete and masonry had to have different crack models to have stable computing. Therefore, it was concluded that for IP calibration, *concrete* and *interface* material properties were the most sensitive. For the OoP wall bend tests, it was both interface and masonry material model. Finally, for the OoP drift driven test it was the combination. However, for the overall response, the governing factor is the concrete material (as for BF, IP calibration) as it is flexural dominated. Yet, to simulate damage to the infill, it was the interface and masonry model (as for OoP bend wall tests).

Collectively, the micromodels can be considered calibrated. It was noted that, when the IP micromodels were calibrated in their direction, only minor adjustments were made to match the OoP tests. Furthermore, when the BF and FI models were calibrated, there

was no need for further calibration of models with openings. Finally, the micromodels were considered calibrated regarding: (a) IP drift loads; (b) OoP drift loads; (c) OoP loads on the infill wall; (d) gravity loads; (e) presence or lack of infill wall; (f) infill walls with window/door openings with different positions.

**Author Contributions:** Conceptualization, F.A., D.P., V.S. and L.A.; methodology, F.A. and D.P.; software, F.A. and D.P.; validation, F.A., D.P. and V.S.; formal analysis, F.A.; investigation, F.A.; resources, F.A., D.P. and L.A.; data curation, F.A. and D.P.; writing—original draft preparation, F.A.; writing—review and editing, F.A., D.P, L.A. and V.S.; visualization, F.A.; supervision, D.P.; project administration, F.A. and D.P.; funding acquisition, F.A. and D.P. All authors have read and agreed to the published version of the manuscript.

**Funding:** This research received no external funding.

**Data Availability Statement:** The data presented in this study are available on request from the corresponding author. The data are not publicly available due to ongoing doctoral dissertation research.

**Acknowledgments:** The research presented in this article forms part of the research project, FRAmed-MAsonry composites for modelling and standardization [HRZZ-IP-2013-11-3013], supported by the Croatian Science Foundation; this support is gratefully acknowledged.

**Conflicts of Interest:** The authors declare no conflict of interest.

## Abbreviations and Symbols

**Latin based**

| | | | |
|---|---|---|---|
| RC | Reinforced concrete | $f_c$ | Compressional strength |
| IP | In-plane | $w_d$ | Plastic displacement |
| OoP | Out-of-plane | $k_{red}$ | Reduction of compressive strength due to cracks |
| MAPE | Mean average percentage error | $G_F$ | Fracture energy |
| FCMC | Fixed crack model coefficient | $K_{nn}$ | Normal stiffness (interface) |
| $A_o$ | Area of opening | $K_{tt}$ | Tangential stiffness (interface) |
| $A_i$ | Area of infill wall | $K_S$ | Area spring stiffness |
| $E$ | Elasticity modulus | **Greek based** | |
| $F$ | Force | $\sigma$ | Normal stress |
| $G_F$ | Fracture energy | $\varepsilon$ | Normal strain |
| $K_{nn}$ | Normal stiffness (interface) | $\beta$ | Direction of plastic flow |
| $K_{tt}$ | Tangential stiffness (interface) | $\mu$ | Poisson's ratio |
| $V$ | Shear force | $\phi$ | Friction coefficient |
| $T_F$ | Friction force | $\tau$ | Shear stress |
| $c$ | Cohesion | $\delta$ | Normal displacement |
| $c_0$ | Initial cohesion | $\nu$ | Shear displacement |
| $c_{ts}$ | Tensile stiffening | $\sigma_y$ | Yield stress |
| $d_r$ | Inter-storey drift ratio | $\sigma_t$ | Ultimate (tensional) stress |
| $f_t$ | Tensional strength | | |

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
