# Peer review of "Development and Calibration of a 3D Micromodel for Evaluation of Masonry Infilled RC Frame Structural Vulnerability to Earthquakes"

_geosciences, doi:10.3390/geosciences11110468_

Round 1

Reviewer 1 Report

The paper is well written and the methodology is clearly exposed. The paper can be published in the present form.

  • To improve the overall quality of the manuscript, the authors should consider to add a section METHODOLOGY where to describe clearly the approach adopted;
  • In the INTRODUCTION, significance and novelty must be clearly stated;

  • State-of-the-art: the authors could also comment on other types of approaches (e.g. Rossi, A., Morandi, P., & Magenes, G. (2021)).

Author Response

The authors thank the reviewers for their interest in our work and for helpful comments that greatly improved the manuscript. The reviewers have brought up some good points and we appreciate the opportunity to clarify our research objectives and results. We have tried to do our best to respond to the points raised. As indicated below, we have checked all the general and specific comments provided by the reviewers and have made necessary changes according to their indications. We have highlighted the changes within the manuscript.

A point-by-point response to the reviewers’ comments and concerns is given in the answers-to-reviewers file.

Reviewer 2 Report

This manuscript presents the development and calibration of 3D micromodels which simulated in-plane and out-of-plane cyclic, quasi-static drift loads on frames with and without infill walls and openings, as well as out-of-plane bend loads on masonry walls. A series of experimental tests was conducted. This is an interesting paper giving insight in the influence of openings within the infill walls in both in- and out-of-plane direction. I highly recommend this paper to be published. I have just only one question.

  • Can you please clarify about the simulation of the contact interface between solids, do you take into account except from sliding (by giving friction properties) and detachment (by giving the ability elements to lose contact and separate from each other)?

Author Response

(The authors gave the same response as above.)

Author Response

(The authors gave the same response as above.)
